# Anti-Inflammatory Activity of *Panax notoginseng* Flower Saponins Quantified Using LC/MS/MS

**DOI:** 10.3390/molecules28052416

**Published:** 2023-03-06

**Authors:** Junchen Liu, Yuehang Wu, Wenrui Ma, Hongyan Zhang, Xianyao Meng, Huirong Zhang, Miaomiao Guo, Xiao Ling, Li Li

**Affiliations:** 1Beijing Key Lab of Plant Resource Research and Development, Institute of Cosmetic Regulatory Science, College of Chemistry and Materials Engineering, Beijing Technology and Business University, Beijing 100048, China; 2Beijing Lan Divine Technology Co., Ltd., Beijing 100048, China

**Keywords:** *Panax notoginseng* flower, UVB, inflammatory factor, antibacterial peptide LL-37, total saponin, LC/MS/MS

## Abstract

*Panax notoginseng* (Burk) F. H. Chen is a traditional Chinese medicinal and edible plant. However, *Panax notoginseng* flower (PNF) is rarely used. Therefore, the purpose of this study was to explore the main saponins and the anti-inflammatory bioactivity of PNF saponins (PNFS). We explored the regulation of cyclooxygenase 2 (COX–2), a key mediator of inflammatory pathways, in human keratinocyte cells treated with PNFS. A cell model of UVB-irradiation-induced inflammation was established to determine the influence of PNFS on inflammatory factors and their relationship with LL–37 expression. An enzyme-linked immunosorbent assay and Western blotting analysis were used to detect the production of inflammatory factors and LL37. Finally, liquid chromatography–tandem mass spectrometry was employed to quantify the main active components (ginsenosides Rb1, Rb2, Rb3, Rc, Rd, Re, Rg1, and notoginsenoside R1) in PNF. The results show that PNFS substantially inhibited COX–2 activity and downregulated the production of inflammatory factors, indicating that they can be used to reduce skin inflammation. PNFS also increased the expression of LL-37. The contents of ginsenosides Rb1, Rb2, Rb3, Rc, and Rd in PNF were much higher than those of Rg1, and notoginsenoside R1. This paper provides data in support of the application of PNF in cosmetics.

## 1. Introduction

*Panax notoginseng* (Burk) F. H. Chen is a widely used traditional Chinese medicinal plant predominantly distributed in Yunnan Province, southwest China. *P. notoginseng* is famous for its therapeutic effects such as blood circulation improvement, blood stasis removal, cardiovascular protection, anti-inflammation, and antioxidation [1]. Although the entire plant has medicinal purposes, including the roots, stems, leaves, and flowers, *Panax notoginseng* flower (PNF) is rarely used, with it primarily being used for making tea. Thus, when planting *P. notoginseng*, only the underground part is collected and the aboveground part is typically discarded, resulting in a significant waste of resources. However, *Panax notoginseng* flower is the part of the whole plant with the highest saponin content, and contains a variety of chemical components such as flavonoids and polysaccharides, which have a wide range of pharmacological effects [2]. In addition, since PNF is much cheaper than the root of *Panax notoginseng* (PNR), this has led to an increasing number of industries developing PNF as an active medicinal ingredient.

PNF can be used to treat hypertension closely related to inflammatory response, chemotherapy stomatitis, pharyngitis, and other inflammatory diseases [3], and it contains many chemical components, including triterpenoids, polysaccharides, flavonoids, and amino acids, with triterpenoids representing the dominant component [1]. Triterpenoids exhibit numerous biological activities [4], and different types of triterpenoid saponins possess different pharmacological properties, including hypoglycemic (ginsenoside Rg1) [5], antioxidative (ginsenoside Re) [6], learning and memory improvement (ginsenoside Rd) [7], neuroprotective (ginsenoside Rb1) [8], immunity enhancement (ginsenoside Rb2) [9], and anti-inflammatory (ginsenoside Rc) [10] properties. Moreover, some saponins are more abundant in PNF than in PNR [11], and their composition can vary among the different plant parts of *P. notoginseng* [12]. The content of protopanaxadiol saponins is higher than that of triol saponins in PNF because the expression of *CYP716A53v2*, which is involved in the conversion of protopanaxadiol to protopanaxatriol, is much lower in the flower than in the root [12].

Skin exposure to ultraviolet B (UVB) rays can cause skin inflammation and induce human immortalized keratinocyte (HaCaT) cells to release inflammatory factors such as prostaglandin E2 (PGE–2), tumor necrosis factor-α (TNF-α), and interleukin-1β (IL-1β) [13]. Among these inflammatory factors, PGE-2 is the major prostanoid produced in human skin. TNF-α plays an important role in photodamage and photoaging and is often induced by UVB irradiation in keratinocytes and dermal fibroblasts [14]. IL–1β is important for innate immunity and is commonly upregulated in inflammatory diseases [15]. IL-1β can also regulate inflammation by inducing the expression of IL-8 and IL-6 [16]. Furthermore, the antibacterial peptide LL-37 is an important contributor to the human innate immune defense system [17] as it exhibits a broad spectrum of antimicrobial activity that can effectively and rapidly protect the host from infection and other diseases caused by microorganisms [18].

Considering research on the anti-inflammatory activity of *Panax notoginseng*, there are few mature cell models in this field. Additionally, most of the studies on its anti-inflammatory activity are carried out for various diseases, such as cardiovascular diseases, nervous system diseases, respiratory system diseases, digestive system diseases, and so on [19]. There are few studies on skin inflammation, and most of them are about the anti-inflammatory effect of saponin monomer. Seol-Hee studied the protective effect of ginsenoside F2 on mice skin inflammation induced by 12-O- tetradecanoylphorbol 13-acetate. Ginsenoside F2 was found to improve skin inflammation induced by 12-O-tetracecanoylphobo-13-acetate by inhibiting the production of IL-17 and ROS in T cells and neutrophils, respectively [20]. Since *Panax notoginseng* saponins have anti-inflammatory activity, we speculated that PNFS, as a complex of various ginsenosides and *Panax notoginseng* saponins, might have better anti-inflammatory activity. Therefore, this study discussed the anti-inflammatory effect of PNF in a UVB-mediated HaCaT cell inflammation model from many aspects, aiming to provide a basis for the application of PNF in the field of cosmetics.

The aim of this study Is to explore the anti-inflammatory bioactivity of saponins derived from PNF (i.e., PNFS) in a cell model of UVB-irradiation-induced inflammation. First, we extract and purify PNFS. Second, we analyze the inhibitory effect of PNFS on COX–2 overproduction to obtain a preliminary understanding of the anti-inflammatory bioactivity of PNF. Third, we induce cell inflammation via UVB irradiation to determine the influence of PNFS on cell inflammatory factors and reveal the anti-inflammatory mechanism of the antibacterial peptide LL-37. Finally, we employ liquid chromatography with tandem mass spectrometry (LC/MS/MS) to quantify the saponin composition of PNFS and confirm the main active ingredients.

## 2. Materials and Methods

### 2.1. Reagents

PNF was acquired from Hebei Jiahenglengbei Co., Ltd., (Yunnan; batch numbers C20071810 and C20071809, respectively). Distilled water was acquired from Guangzhou Watsons Food & Beverage Co., Ltd. D101 macroporous resin, standard saponins (ginsenosides Rb1, Rb2, Rb3, Rc, Rd, Re, Rg1, and notoginsenoside R1), and vanillin were acquired from Shanghai Yuanye Biotechnology Co., Ltd. (batch numbers B21053, B21054, B21055, B21099, B210056, B21050, B21051, and B21052, respectively). Anhydrous ethanol and glacial acetic acid were acquired from Fuchen Chemical Reagent Co., Ltd. (Tianjin, China), and chromatographic grade methanol and acetonitrile were acquired from Aladdin Reagent Co., Ltd. (Shanghai, China). Dulbecco’s modified Eagle’s medium, heat-inactivated fetal bovine serum, Dulbecco’s phosphate-buffered saline (PBS), penicillin–streptomycin, and trypsin 2.5% were acquired from Life Technologies, Inc. (Grand Island, NY, USA). β-actin, PGE-2, IL-1β, and TNF-α antibodies were acquired from Cell Signaling Technology (Beverly, MA, USA). PGE-2, IL-1β, and TNF-α enzyme-linked immunosorbent assay (ELISA) kits were purchased from BioChell (Shanghai, China). The COX-2 inhibitor screening kit was acquired from Beyotime Biotechnology (Shanghai, China).

### 2.2. Extraction and Purification of PNFS

Extraction and purification of *Panax notoginseng* flower were carried out according to the improved experimental method of Ye Hui et al. [21]. After crushing, 50 g of PNF was added to 250 mL of 75% ethanol solution (ethanol solution containing 25% aqueous solvent), and ultrasonic extraction was performed at 60 °C (power: 560 W) for 30 min. The filtrate was then extracted and collected. After three extractions, the 75% ethanol extract was collected and concentrated at 60 °C. The saponins from the 75% ethanol extract were then enriched with D101 macroporous resin by conducting sample loading to D101 macroporous resin at 1:20. Water was then used to wash away impurities, and 50% ethanol was used to elute the total saponins in the 75% ethanol extract. The purified PNF extract was then obtained and used as PNFS in the subsequent anti-inflammatory assay. The total saponin content in the ethanol extract and PNFS were then measured as described in Section 2.3.

### 2.3. Measurement of Total Saponin Content

Ginsenoside Re was used to represent total saponins to establish a linear relationship between the total saponin mass and absorbance value. The ginsenoside Re standard solution was diluted to five different concentrations to obtain the calibration curve. Then, 1 mL of ginsenoside Re standard solution at each concentration was used to measure the absorbance value. After drying the standard solution drying, we added 0.2 mL of 5% vanillin glacial acetic acid solution (5 g vanillin with glacial acetic acid to 100 mL), mixed the solution thoroughly, then added 0.8 mL of perchloric acid and mixed again. The mixture was then heated in a 60 °C water bath for 10 min then cooled with ice water, and 5.0 mL of glacial acetic acid was added. Then, 0.2 mL of the mixture was added to the 96-well plate, and the absorbance was measured at 560 nm using a microplate reader [22]. Finally, 1 mg of lyophilized powder of the 75% ethanol extract and purified extract (PNFS) were used to determine the total saponin content.

### 2.4. Measurement of COX-2 Overproduction

All reagents in the COX-2 inhibitor screening kit (except rhCOX-2) were placed at 28 °C for 30 min, centrifuged to obtain the precipitate, mixed again, then set aside. COX-2 probe, COX-2 Cofactor, COX-2 substrate, and PNFS were all dissolved in dimethyl sulfoxide (DMSO). COX-2 Cofactor and rhCOX-2 were diluted with COX-2 Assay Buffer at ratios of 1:49 and 1:24, respectively. The COX-2 substrate was mixed thoroughly with a substrate buffer solution of the same volume, and the mixture was diluted with pure water at a ratio of 1:24. The positive control inhibitor celecoxib was diluted to the appropriate concentration with DMSO. A 96-well plate was used for the detection. The blank control group contained 80 μL of COX-2 Assay Buffer, 5 μL of COX-2 Cofactor working solution, and 5 μL of DMSO. The 100% enzyme activity control group contained 75 μL of COX-2 Assay Buffer, 5 μL of COX-2 Cofactor working solution, 5μL of COX-2 working solution, and 5 μL of DMSO. The positive inhibitor control group contained 75 μL of COX-2 assay buffer, 5 μL of COX-2 Cofactor working solution, 5 μL of COX-2 working solution, 5 μL of DMSO, and 5 μL of celecoxib. The experimental group contained 75 μL of COX-2 Assay Buffer, 5 μL of COX-2 Cofactor working solution, 5 μL of COX-2 working solution, 5 μL of DMSO, and 5 μL of PNFS. All reagents were mixed thoroughly and incubated at 37 °C for 10 min. Subsequently, 5 μL of COX-2 Probe and 5 μL of COX-2 substrate were added to each well. Fluorescence was determined after incubation at 37 °C for 5 min in the absence of light. The excitation wavelength was 560 nm and the emission wavelength was 590 nm. The percentage inhibition of each well was calculated using the following equation: inhibition rate (%) = (F_1_ − F_2_)/(F_1_ − F_3_) * 100%, where F_1_, F_2_, and F_3_ represent the fluorescence value of the 100% enzyme activity control group, the experimental group, and the blank control group, respectively.

### 2.5. Cell Culture

HaCaT cells were cultured in Dulbecco’s modified Eagle’s medium with 10% fetal bovine serum containing dual antibiotics (1% penicillin and 1% streptomycin). The cells were incubated at 37 °C in a 5% CO_2_ atmosphere then used for subsequent assays when the cultured cells reached approximately 80% confluence.

### 2.6. Cell Viability

Cell viability was evaluated using the cell counting kit-8 (CCK-8) assay. To determine the optimal concentrations of PNFS for measurement of anti-inflammatory factor overproduction, HaCaT cells were pretreated with various concentrations of PNFS. After incubation for 24 h, 10 μL of CCK-8 solution was added, and the cells were incubated at 37 °C for 1 h. To obtain the percentage of viable cells, the absorbance of each treatment was measured using a microplate reader at a wavelength of 450 nm. To determine the optimal dose of UVB irradiation, HaCaT cells were irradiated with various doses of UVB, and cell viability was evaluated according to the aforementioned method. The irradiation intensity of UVB was monitored with a 4006 B ultraviolet phototherapy radiometer and an SUV6 photodetector.

### 2.7. Measurement of PGE-2, IL-1β, TNF-α, and LL-37 Overproduction by ELISA 

An ELISA kit was used to analyze the UVB-irradiation-induced inflammation model, the anti-inflammatory factor bioactivity of PNFS, and the influence of PNFS on the expression of LL-37 in HaCaT cells. To validate the UVB-induced inflammation model, PGE-2 overproduction induced by UVB was measured using an ELISA kit. For the anti-inflammatory factor assay, HaCaT cells were divided into a blank control group, a UVB irradiation group, three experimental groups (with three concentrations of PNFS obtained from the pretreatment experiment in Section 2.6), and a positive control group (dexamethasone at a mass fraction of 0.01%). HaCaT cells (6 × 10^5^ cells per well) were grown in 6-well plates. Before UVB irradiation, cells were washed with 1 mL of PBS and replaced with 1 mL of fresh PBS in each well. Cells were irradiated without a plastic dish lid at the desired intensity, which was obtained from the experiment in Section 2.6. To detect the inhibitory effects of PNFS on PGE-2, IL-1β, and TNF-α produced by UVB irradiation, HaCaT cells were grown in the 6-well plates and either treated with PNFS at different concentrations or untreated for 24 h after UVB irradiation. Then, the culture supernatants were collected and measured with corresponding ELISA kits to estimate the concentrations of PGE-2, IL-1β, and TNF-α. The expression level of LL-37 was subsequently measured in HaCaT cells with the corresponding ELISA kit after 24 h of treatment with PNFS.

### 2.8. Western Blotting Analysis 

Western blotting analysis was used to verify the UVB-irradiation-induced inflammation model, the anti-inflammatory factor bioactivity of PNFS, and the influence of PNFS on the expression of LL-37 in HaCaT cells. β-Actin was used as an internal reference [23]. HaCaT cells were detached from the culture plates by trypsinization, stored at 0 °C, and washed with ice-cold PBS. The cells were lysed with lysis buffer then centrifuged in a refrigerated centrifuge at 4 °C for 10 min to collect total cell proteins. The total protein content was quantified using a BCA protein assay kit. SDS-PAGE (10%) was used to separate the proteins, and the proteins to be tested were electrotransferred onto a polyvinylidene difluoride membrane. A 5% (*w*/*v*) bovine serum albumin solution was used to block the membrane for 1 h, after which the membrane was incubated with primary antibodies (anti-PGE-2, anti-IL-1β, anti-TNF-α, and anti-β-actin) at 4 °C for 12 h then incubated with the corresponding secondary antibodies at room temperature for 1 h. Enhanced chemiluminescence was used to assess the number of bound antibodies. The relative expression levels of target proteins were calculated based on the optical density of the electrophoresis bands compared with β-actin. The relative expression level of LL-37 was subsequently measured using the aforementioned method with the corresponding antibody.

### 2.9. Quantitation of Saponins in PNFS by UHPLC/MS/MS

Ultra-high performance (UHPLC) LC/MS/MS was performed using an ACQUITY UHPLC system with a BEH C18 column (2.1 × 50 mm, 1.7 μm) equipped with an XEVO TQD triple quadruple mass spectrometer and an electrospray ionization source for chromatographic analysis of ginsenosides and notoginsenosides. The column temperature was maintained at 25 °C and the flow rate was set to 0.4 mL/min. The autosampler was conditioned at 10 °C and the injection volume was 10 μL. Detection was performed in multiple reaction monitoring mode, with a dwell time of 200 ms. Water (A) and methanol (B) gradient elution were used for all quantitation. The gradient program was as follows: initial, 30% B; 0.00–7.50 min, 30–64% B; 7.5–18.00 min, 64–72% B; 18.00–18.10 min, 72–30% B; and 18.10–19.00 min, 30% B. To obtain appropriate calibration curves, these standard solutions were diluted to different concentrations for quantitation; each calibration curve was performed with four concentrations of the standard solutions. The lyophilized powder of PNF was diluted to different multiples for the measurement of different ginsenosides and notoginsenosides. The source parameters after optimization by MS tone and IntelliStart are shown in Table 1 and Table 2. The MRM chromatogram of each compound is shown in Figure 1.

### 2.10. Statistical Analysis

Figures were analyzed using GraphPad 5.0 software. The significance between each group was analyzed by one-way ANOVA analysis of variance, with values only considered statistically significant when *p* < 0.05. Image J software was used to quantify protein levels on the Western blots. All experiments were repeated at least three times, and the results are presented as mean values ± the standard deviation. The limit of detection (LOD) and limit of quantitation (LOQ) for the tested ginsenosides were estimated at signal-to-noise ratios of 3 and 10, respectively, by injecting a series of diluted solutions with known concentrations. The precision of the extracted ion chromatogram peak area measurements for the eight ginsenosides were calculated as the relative standard deviations (RSDs) of six repeated runs. The sample stability was monitored by analyzing the same sample solution every 6 h for 36 h. The RSDs of the eight ginsenoside measurements were regarded as indices of the stability of the analytical system. The reproducibility of the method was evaluated by running six replicate samples of PNF prepared independently on a single day. The recovery of ginsenosides was measured by standard addition methods within the same day. The recovery results were calculated as the standard solution of mixed ginsenosides with three concentration levels, that is, approximately 50%, 100%, and 150% of the target component added to the standard ginsenosides at known concentrations. 

## 3. Results and Discussion

### 3.1. Total Saponin Content in 75% Ethanol Extract and Purified Extract (PNFS)

In this study, 50 g of PNF was extracted with 75% ethanol to obtain 22.04 g of extract, which exhibited a total saponin content of 32.58%, calculated using the calibration curve shown in Figure 2. Then, the 9.85 g extract was adsorbed with 50% ethanol by 197 g macroporous resin to obtain 4.40 g of eluted solid, in which the total saponins content was 62% and the enriched saponins mass was 2.73 g. The results indicate the validity of the extraction and purification methods, which increased the total saponin content by approximately two-fold.

### 3.2. Effect of PNFS on COX-2 

Pro-inflammatory enzymes produced by UVB exposure and the subsequent activation of associated signaling pathways, such as COX-2, trigger the secretion of specific inflammatory mediators, including prostaglandins and various cytokines [24]. UVB-induced COX-2 expression also plays a major role in UVB-induced PGE-2 production, inflammation, and keratinocyte proliferation [25]. In this study, PNFS clearly inhibited COX-2 activity at concentrations of 100, 200, 500, and 1000 μg/mL in a concentration-dependent manner (Table 3). Previous studies have shown that both inhibited COX-2 activity and reduced COX-2 expression significantly reduce UVB-induced skin cancer, whereas the overexpression of COX-2 significantly increases UVB-induced skin cancer [18], indicating that PNFS can improve skin inflammation and prevent skin cancer from UVB irradiation by inhibiting COX-2. 

### 3.3. Cell Viability and Development of the UVB-Irradiation-Induced Inflammation Model

As shown in Table 4, HaCaT cells treated with PNFS at 50, 100, and 200 μg/mL exhibited high cell viability, with a cell survival rate above 80%. Therefore, we selected these PNFS concentrations for subsequent experiments to avoid impacting the cell viability and to ensure successful experiments. As shown in Figure 3, the survival rate of HaCaT cells decreased with an increase in UVB irradiation intensity. The cell survival rates of the 50, 75, and 100 mJ/cm^2^ irradiation groups were not significantly different, whereas those of the 125 mJ/cm^2^ and 150 mJ/cm^2^ groups were significantly lower, indicating that UVB intensity of greater than 100 mJ/cm^2^ significantly reduced the cell survival rate. The secretion level of PGE-2 in HaCaT cells irradiated with different doses of UVB was detected using the corresponding ELISA kit. As shown in Figure 3, the secretion of PGE-2 in HaCaT cells increased with an increase in UVB intensity. At an intensity of less than 25 mJ/cm^2^, there was no significant difference in PGE-2 secretion in HaCaT cells from that of the blank control group. Although PGE-2 secretion in the 50, 75, 100, 125, and 150 mJ/cm^2^ groups was significantly different from that in the blank control group, and there was a significant difference between UVB irradiation at 75 mJ/cm^2^ and 100 mJ/cm^2^, there was no significant difference between UVB irradiation at 100 mJ/cm^2^ and 125 mJ/cm^2^. The Western blotting results revealed that 100 mJ/cm^2^ of UVB irradiation led to PGE-2 overproduction in HaCaT cells, thereby inducing inflammation (Figure 4). Thus, 100 mJ/cm^2^ of UVB irradiation was chosen in the following assay.

### 3.4. Effect of PNFS on PGE-2, IL-1β, and TNF-α Determined by ELISA

Exposure of HaCaT cells to UVB irradiation for 24 h resulted in increased levels of PGE-2, IL-1β, and TNF-α. As PNFS reduced the expression of COX-2, it follows that PNFS also downregulated the production of PGE-2 because PGE-2 is synthesized via the sequential activities of phospholipase A2, COX, and PGE synthase. Indeed, as shown in Figure 5, the PNFS treatment groups reduced the expression of PGE-2, IL-1β, and TNF-α in a concentration-dependent manner. This is consistent with the results of Hyo-Won Jung et al. [26]. Specifically, a PNFS concentration of 200 μg/mL downregulated the production of IL-1β and TNF-α to close to that of the control group.

### 3.5. Effect of PNFS on PGE-2, IL-1β, and TNF-α Determined by Western Blotting

Western blotting was used to detect the secretion of PGE-2, IL-1β, and TNF-α in HaCaT cells. As shown in Figure 6, PNFS exhibited inhibitory effects on the expression of all three inflammatory factors. PNFS concentrations of 100 μg/mL and 200 μg/mL significantly inhibited the expression of inflammatory factors (*p* < 0.01). Thus, the Western blotting results were consistent with the ELISA results. Several previous reports have demonstrated that PGE-2 can increase UVB-induced skin inflammation and tumorigenesis, IL-1β can upregulate the production of matrix-degrading metalloproteinases (MMPs), which can cause photoaging, and TNF-α can promote apoptosis, lymphocyte activation, and hyperproliferative skin disorders. Therefore, the observed reduction in these inflammatory factors in HaCaT cells treated by PNFS treatment will not only alleviate skin inflammation but also prevent other skin damage caused by UVB irradiation, such as photoaging.

### 3.6. Effect of PNFS on LL-37 Expression

Compared with the blank control group, secretion of the antibacterial peptide LL-37 was significantly increased by PNFS at different concentrations (Figure 7 and Figure 8). Western blotting was performed to verify the results, which confirmed that PNFS promoted the expression of the antibacterial peptide LL-37. According to our previous results showing that PNFS has certain inhibitory effects on UVB-induced inflammatory factors, it is likely that some interactions exist between these inflammatory factors and LL-37. Indeed, multiple studies have demonstrated that LL-37 regulates skin inflammation through multiple signaling pathways. Specifically, LL-37 can reduce the expression of TNF-α at the transcriptional level; however, it also induces COX-2-dependent induction of PGE-2 in keratinocytes and works in synergy with IL-1β to reinforce certain innate immune responses [27]. Our results indicate that PNFS directly decreased the production of inflammatory factors. However, according to the existing literature, the production of COX-2 and PGE-2 should be upregulated under the effect of LL-37, which does not agree with our experimental findings. Therefore, we speculate that the downregulation of COX-2 and PGE-2 by PNFS was stronger than that of LL-37. The synergistic interaction between IL-1β and LL-37 was also likely weakened to a certain extent.

Studies have shown that in people with Atopic Dermatitis (AD), the levels of inflammatory cytokines TNF-α, PEG-2 and IL-1β are increased [28], and the expression of antimicrobial peptide LL-37 is decreased, which leads to the outbreak of inflammation. The PNFS can increase the expression of antibacterial peptide LL-37 and reduce inflammatory factors. So, it can participate in the inflammatory regulation process of AD, dry sensitive people and people with damaged skin barrier, which is conducive to the development of self-repair of inflammation.

### 3.7. Major Saponin Contents in PNFS

To validate the UHPLC/MS/MS method, calibration curves of each compound were established by plotting the peak area (y) versus the concentration (x), as shown in Table 5. The LOD, LOQ, precision RSD, and repeatability of all calibration curves are summarized in Table 5. All calibration curves exhibited good linearity (r^2^ > 0.99), and the results (RSD < 2.5%) showed that the sample solutions were stable during quantitation. To assess repeatability, the standard samples were redissolved and analyzed again after establishing the calibration curves. The results indicated good repeatability of the UHPLC/MS/MS method for quantifying ginsenoside and notoginsenoside contents in PNF. The saponin concentration was then calculated based on the calibration curves.

The quantitation results show that Rb3 was most abundant in PNFS, followed by Rc and Rb2, all of which are protopanoxadiol-type ginsenosides (Table 6). Re was the least abundant in PNFS, followed by Rg1 and NG R1. Thus, according to LC/MS/MS, the ginsenosides Rb2, Rb3, and Rc were the main components of PNFS, which all exhibit different bioactivities in the skin. Rb2 accelerates cell proliferation, the expression of proliferation-related factors, and epidermal formation, which promotes skin injury healing [29]. Rb3 exhibits potential skin anti-photoaging effects [30], and not only suppresses UVB-induced intracellular ROS levels and MMP-2 and MMP-9 secretion, but also enhances UVB-depleted total glutathione levels and superoxide dismutase activity. Rc treatment can also inhibit UVB-induced ROS generation and prevent UVB-induced glutathione depletion and superoxide dismutase activity suppression, as well as downregulate the expression and activity of MMP-2 and MMP-9 [31]. Thus, PNFS can not only improve skin inflammation, but also promote skin healing and prevent skin aging and oxidative damage caused by UVB exposure.

We conclude that the ginsenosides Rb2, Rb3, and Rc are the main bioactive components of PNFS that decrease the production of COX-2, PGE-2, IL-1β, and TNF-α, and increase the production of LL-37 in the UVB-induced inflammation model established in this study. It remains unclear whether the remaining saponins play a role in the anti-inflammatory bioactivity of PNFS; therefore, this aspect should be investigated further.

## 4. Conclusions

In this study, we verified that the purified extract of total saponins from *P. notoginseng* flowers is capable of inhibiting inflammation in the skin. PNFS at a concentration of 200 μg/mL downregulated the production of the inflammatory factors IL-1β and TNF-α to close to that of the control group in the UVB-induced inflammation model, indicating that PNFS can improve skin inflammation. PNFS also increased the expression of the antibacterial peptide LL-37, indicating that PNFS downregulated the expression levels of inflammatory factors by upregulating the expression level of LL-37. In addition, UHPLC/MS/MS showed that the main ginsenosides in PNF were Rb1, Rb2, Rb3, Rc, and Rd, and the specific mechanism of action needs further discussion. In summary, PNFS exhibited excellent COX-2 inhibition and anti-inflammatory ability. Thus, we suggest developing this extract as a natural plant resource that can reduce skin inflammation caused by UVB irradiation with fewer adverse effects.

## Figures and Tables

**Figure 1 molecules-28-02416-f001:**
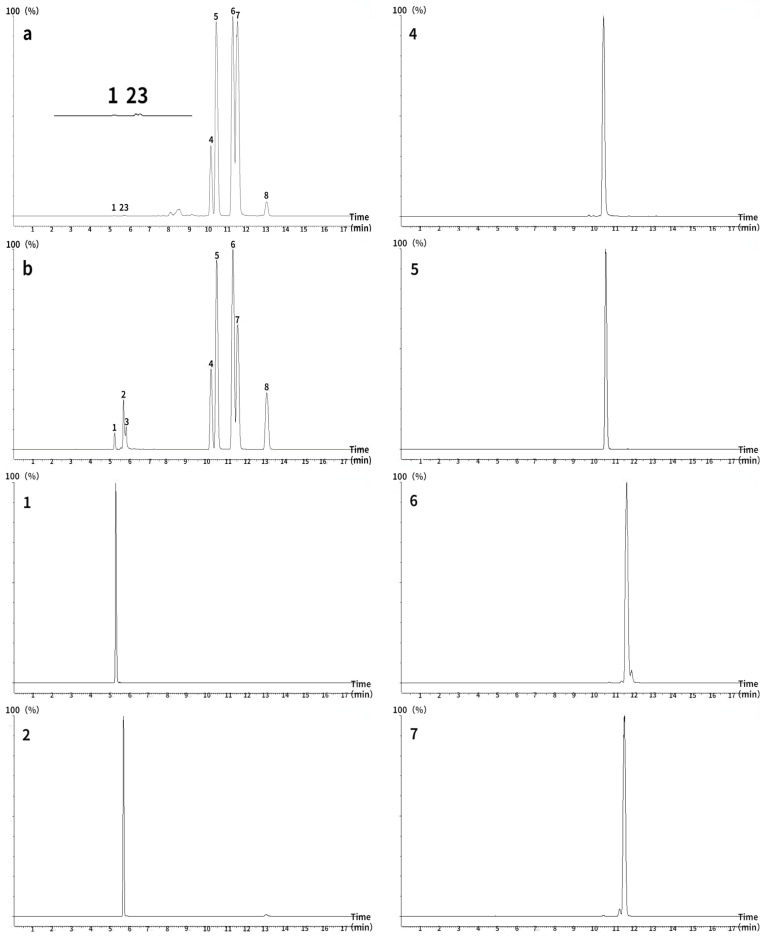
Total ion chromatograms of eight ginsenosides measured using ultra-high performance liquid chromatography–tandem mass spectrometry (ACQUITY UHPLC BEH C18 column; 2.1 × 50 mm, 1.7 μm). (**a**) *Panax notoginseng* flower, (**b**) standard. 1: NG R1, 2: Re, 3: Rg1, 4: Rb1, 5: Rc, 6: Rb2, 7: Rb3, 8: Rd.

**Figure 2 molecules-28-02416-f002:**
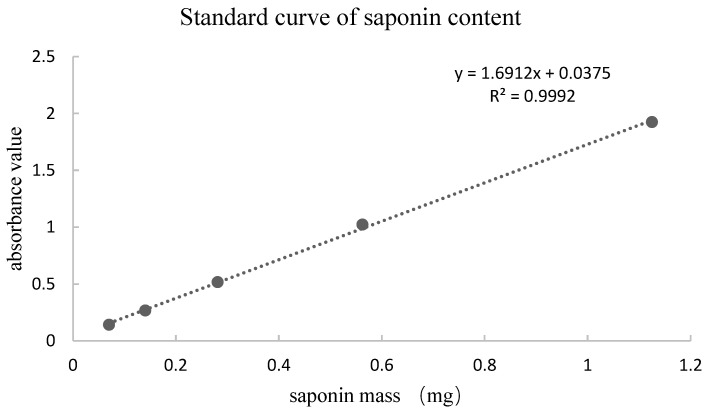
Standard curve of ginsenoside Re standard solution.

**Figure 3 molecules-28-02416-f003:**
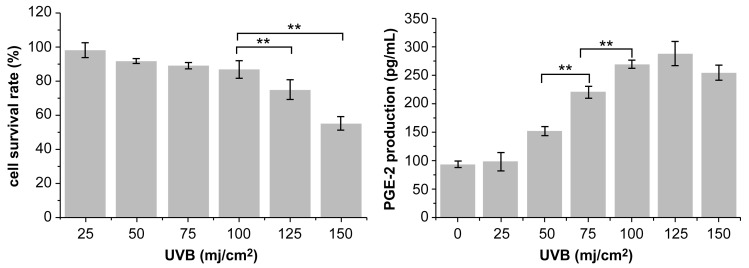
Effect of UVB irradiation intensity on cell viability and PGE-2 production. ** *p* < 0.01. UVB: Ultra-violet B, PGE-2: prostaglandin E2.

**Figure 4 molecules-28-02416-f004:**
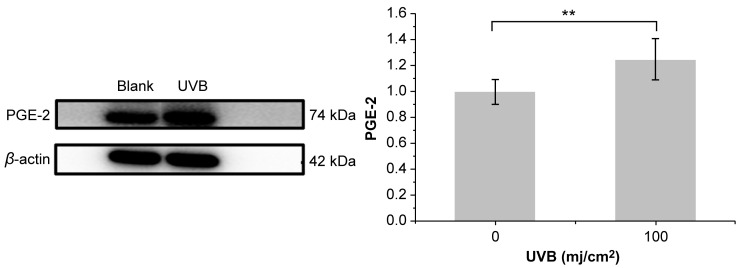
Effect of 100 mJ/cm^2^ of UVB irradiation on PGE-2 production. ** *p* < 0.01. UVB: Ultra-violet B, PGE-2: prostaglandin E2.

**Figure 5 molecules-28-02416-f005:**
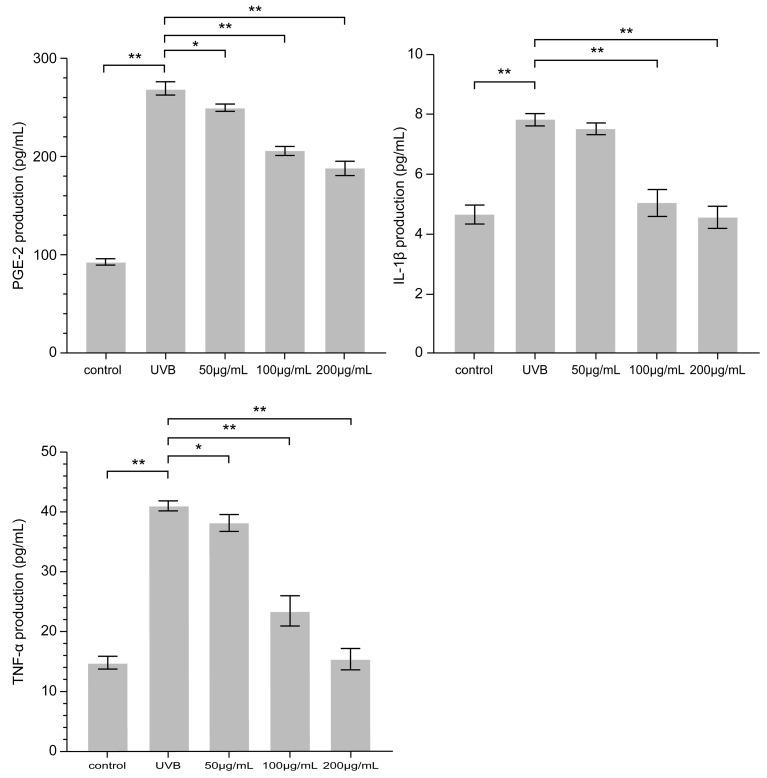
Effect of PNFS on PGE-2, IL-1β, and TNF-α production determined using ELISA. * *p* < 0.05, ** *p* < 0.01. PNFS: *Panax notoginseng* flower saponins, PGE-2: prostaglandin E2, IL-1β: interleukin-1β, TNF-α: tumor necrosis factor-α, ELISA: enzyme-linked immunosorbent assay.

**Figure 6 molecules-28-02416-f006:**
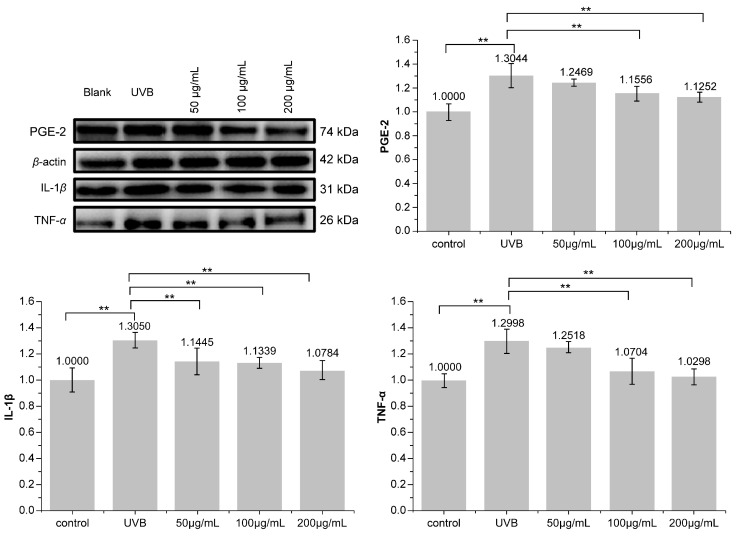
Effect of PNFS on PGE-2, IL-1β, and TNF-α production determined using Western blotting. ** *p* < 0.01. PNFS: *Panax notoginseng* flower saponins, PGE-2: prostaglandin E2, IL-1β: interleukin-1β, TNF-α: tumor necrosis factor-α.

**Figure 7 molecules-28-02416-f007:**
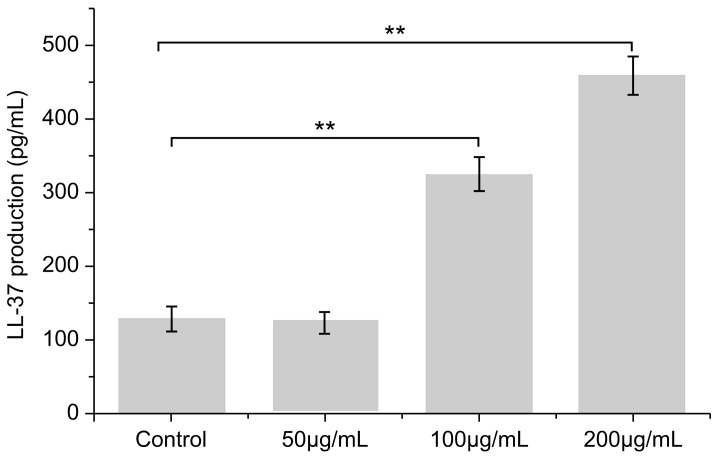
Effect of PNFS on peptide LL-37 production determined using ELISA. ** *p* < 0.01. PNFS: *Panax notoginseng* flower saponins.

**Figure 8 molecules-28-02416-f008:**
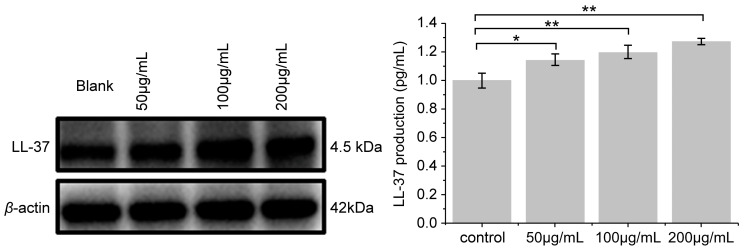
Effect of PNFS on LL-37 production determined using Western blotting. * *p* < 0.05, ** *p* < 0.01. PNFS: *Panax notoginseng* flower saponins.

**Table 1 molecules-28-02416-t001:** Mass spectrometric conditions of the eight ginsenosides measured in this study.

Conditions	Parameter
Capillary (kV)	4
Cone (V)	100
Desolvation temperature (°C)	450
Desolvation (L/h)	700
Cone (L/h)	30

**Table 2 molecules-28-02416-t002:** Detected ion pairs of the eight ginsenosides measured in this study.

Ginsenoside	Adduct Formula	Monoisotopic Mass	Parent (*m*/*z*)	Daughter (*m*/*z*)
Rb1	[M + Na]^+^	1109.31	1131.59	335.11
Rb2	[M + Na]^+^	1079.27	1101.65	335.04
Rb3	[M + Na]^+^	1079.27	1101.59	365.08
Rc	[M + Na]^+^	1079.27	1101.46	334.98
Rd	[M + Na]^+^	947.15	969.39	789.36
Re	[M + Na]^+^	947.16	969.46	789.35
Rg1	[M + Na]^+^	800.49	823.50	643.32
NG R1	[M + Na]^+^	933.13	955.59	203.04

**Table 3 molecules-28-02416-t003:** Effect of *Panax notoginseng* flower saponins (PNFS) concentrations on cyclooxygenase 2 (COX-2).

Concentration (μg/mL)	Blank Control	Positive Control	100	200	500	1000
Inhibition ratio (%)	22.58 ± 1.22	92.01 ± 4.49	46.86 ± 1.37 **	52.11 ± 5.02 **	65.50 ± 4.33 **	82.41 ± 3.56 **

The statistical significance was compared between the PNFS and blank control groups. ** *p* < 0.01.

**Table 4 molecules-28-02416-t004:** Effect of PNFS concentrations on cell viability. PNFS: *Panax notoginseng* flower saponins.

Concentration (μg/mL)	50	100	200	500	1000
Cell survival Rate (%)	97.80 ± 1.89	92.14 ± 6.09	84.26 ± 3.35	83.87 ± 2.47	77.64 ± 4.56

**Table 5 molecules-28-02416-t005:** Linear range, limit of detection (LOD), limit of quantitation (LOQ), relative standard deviations (RSDs) of precision, stability, repeatability, and recovery of the eight ginsenosides analyzed in this study.

Compound	Calibration Curve	r^2^	Linear Ranges (μg/mL)	LOQ (ng/mL)	LOD (ng/mL)	Precision RSD (%)	Stability RSD (%)	Repeatability RSD (%)	Recovery (%)
**Rb1**	Y = 50.1255x − 3713.21	0.9926	0.1–100	0.004	0.002	3.45	4.22	2.51	96.21
**Rb2**	Y = 53.2646x − 4165.20	0.9911	0.1–100	0.045	0.025	2.37	4.53	3.21	94.43
**Rb3**	Y = 30.9952x − 3740.43	0.9999	0.1–100	0.010	0.005	3.55	4.12	3.76	103.76
**Rc**	Y = 37.1653x + 1932.67	0.9980	0.1–100	2	0.5	1.28	3.66	4.19	95.64
**Rd**	Y = 7.1643x + 940.04	0.9970	0.1–100	1	0.2	2.32	3.76	4.72	92.57
**Re**	Y = 8.5135x + 3545.47	0.9984	0.1–100	1	0.5	3.97	3.42	3.11	97.49
**Rg1**	Y = 3.2125x + 3114.94	0.9991	0.1–100	1.5	0.6	2.43	4.51	2.57	103.52
**NG R1**	Y = 2.9907x + 210.58	0.9975	0.1–100	2	0.8	3.64	3.73	3.96	106.75

**Table 6 molecules-28-02416-t006:** Comparison of ginsenoside and notoginsenoside contents in *Panax notoginseng* flower (n = 3).

No.	Ginsenoside	PNF (μg/mg)
**1**	**NG R1**	1.98 ± 0.38
**2**	**Re**	1.62 ± 0.36
**3**	**Rg1**	2.78 ± 0.58
**4**	**Rb1**	83.06 ± 1.24
**5**	**Rc**	163.66 ± 0.63
**6**	**Rb2**	108.53 ± 1.45
**7**	**Rb3**	223.40 ± 0.22
**8**	**Rd**	79.42 ± 0.53

## Data Availability

The data presented in this study are available on request from the corresponding author.

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
