# Peer review of "Anti-Inflammatory Activity of *Panax notoginseng* Flower Saponins Quantified Using LC/MS/MS"

_molecules, 2023, doi:10.3390/molecules28052416_

Round 1

Reviewer 1 Report

In this research, the extraction, and synthesis of ginsenoside, and exploring the research progress of ginsenoside in the field of anti-inflammatory activity has been demonstrated. The knowledge provides a good foundation for the application and development of Panax notoginseng flower saponins in the future.

Overall, the manuscript's ideas appear justified. However, the results of anti-inflammatory activity should be compared with some previous publications. Listed are some comments regarding the submitted manuscript.

Line 40: It is better if the manuscript provides the data (or examples) indicating that researchers have increasingly begun to recognize the bioactivity of PNF.

Line 56: What’s “UVB radiation”? UVB → Ultraviolet B (UVB)

Line 67: panax notoginseng → Panax notoginseng

Line 91, Materials and Methods: Please provide the references for the methods section.

Author Response

Response to Reviewer 1 Comments

Thank you very much for taking time out of your busy schedule to review this paper. The author has revised it one by one according to your comments and marked it in yellow. The specific reply is as follows:

Point 1: Overall, the manuscript's ideas appear justified. However, the results of anti-inflammatory activity should be compared with some previous publications.

Response 1: Thank you for your comments. The comparison of the anti-inflammatory results of Panax Notoginseng flower with Hyo-Won Jung et al was added in line 310, and reference 29 was introduced.

Point 2: Line 40: It is better if the manuscript provides the data (or examples) indicating that researchers have increasingly begun to recognize the bioactivity of PNF.

Response 2: Thank you for your comments. The use of “ However, Panax notoginseng flower is the part of the whole plant with high saponin content, and contains a variety of chemical components such as flavonoids and polysaccharides, which has a wide range of pharmacological effects. In addition, PNF is much cheaper than the root of Panax notoginseng (PNR) ” instead of “ However, researchers have increasingly begun to recognize the bioactivity of PNF, which, along with the fact that PNF is much cheaper than the root of P. notoginseng (PNR) ”.

Point 3: Line 56: What’s “UVB radiation”? UVB → Ultraviolet B (UVB).

Response 3: Thank you for your comments. The use of “ Skin exposure to ultraviolet B (UVB) rays ” instead of “ UVB radiation ”.

Point 4: Line 67: panax notoginseng →Panax notoginseng.

Response 4: Thank you for your comments. “ panax notoginseng ” has been changed to “ Panax notoginseng ” in this article.

Point 5: Line 91, Materials and Methods: Please provide the references for the methods section.

Response 5: Thank you for your comments. Reference materials for the experiment of Panax notoginseng flower extraction are introduced in 2.2. Reference materials for western blotting analysis are introduced in 2.8.

Reviewer 2 Report

This manuscript entitled “Anti-inflammatory activity of Panax notoginseng flower saponins quantified using LC/MS/MS” by Liu et al., studied the quantification of ginsenosides from Panax notoginseng flower using LC-MS-MS and evaluated their Anti-inflammatory activity. There are some issues with this manuscript which must be revised before it gets accepted for publication.

- There are so many typographical mistakes which must be carefully rectified.

- Extraction and purification of PNFS: authors say 75% 75% ethanol solution what is the other 25% solvent??? Please rectify the sentence for more clarity.

-In fig.1 chromatogram (a) Panax notoginseng flower, First three compounds i.e. compounds 1, 2 and 3 are not seen in chromatogram so authors should report only 5 compounds i.e. compounds 4,5,6,7 and 8 (RT 10-14 min).

-Authors should improve the introduction section explaining importance of natural products by including some new references such as

Benzofurans and sterol from the seeds of Styrax obassia

Chemistry of natural compounds 44 (4), 435-439, 2008

Antibiotic and Antibiofilm Activities of Salvadora persica L. Essential Oils against Streptococcus mutans: A Detailed Comparative Study with Chlorhexidine.

Pathogens 9 (1), 66 , 2020

Author Response

Response to Reviewer 2 Comments

Thank you very much for taking time out of your busy schedule to review this paper. The author has revised it one by one according to your comments and marked it in yellow. The specific reply is as follows:

Point 1: There are so many typographical mistakes which must be carefully rectified.

Response 1: Thank you for your comments. The typographical error has been corrected. UVB → Ultraviolet B (UVB). panax notoginseng →Panax notoginseng

Point 2: Extraction and purification of PNFS: authors say 75% 75% ethanol solution what is the other 25% solvent??? Please rectify the sentence for more clarity.

Response 2: Thank you for your comments. In line 114, 75% ethanol solution was changed to 75% ethanol solution (ethanol solution containing 25% aqueous solvent).

Point 3: In fig.1 chromatogram (a) Panax notoginseng flower, First three compounds i.e. compounds 1, 2 and 3 are not seen in chromatogram so authors should report only 5 compounds i.e. compounds 4,5,6,7 and 8 (RT 10-14 min).

Response 3: Thank you for your comments. The chromatogram contains peaks of compounds 1,2,3, but they are not obvious enough. The authors have amplified the three peaks.

Point 4: Authors should improve the introduction section explaining importance of natural products by including some new references.

Response 4: Thank you for your comments. The author has added the pharmacological effects of Panax Notoginseng flower in the introduction and cited references [3] to illustrate the importance of natural products.

Round 2

Reviewer 2 Report

The Manuscript has been improved now, however, there are still some important information need to be incorporated

-Figure 1 shows that peak number 1 is the lowest one howver, the quantity of this compound in Table 6 is not matching why???  

-In Table 6 compound number of each compounds should be provided as they have been given in Figure 1

-From 50g PNF how much aqueous ethanoilc extract was obtained??? Weight of extract should be provided.

-Authors should provide the weight of PNF extract and D101 119 macroporous resin used for the enrichment.

-weight of total saponins from 75% aqueous ethanolic extract after 50% ethanolic elution should be mentioned.

-There are still so many English typographical mistakes which needs to be taken care of.

Author Response

Response to Reviewer 2 Comments

Thank you very much for taking time out of your busy schedule to review this paper. The author has revised it one by one according to your comments and marked it in yellow. The specific reply is as follows:

Point 1: Figure 1 shows that peak number 1 is the lowest one. However, the quantity of this compound in Table 6 is not matching why???

Response 1: Thank you for your comments. The response degree of each saponin in lc-ms/ms is different, so the standard curve is different (the formula of specific standard curve is in Table 5). Its content is brought into the standard curve of each saponin according to the peak area obtained in lc-ms/ms, and the final content is calculated by lc-ms/ms Therefore, Saponin 1 was the least responsive but not the least abundant.

Point 2: In Table 6 compound number of each compounds should be provided as they have been given in Figure 1.

Response 2: Thank you for your comments. Its compound numbers have been added in Table 6 and correspond to Figure 1.

Point 3: From 50g PNF how much aqueous ethanoilc extract was obtained??? Weight of extract should be provided.

Response 3: Thank you for your comments. In this paper, 22.04g of extract obtained from 50g of PNF was added in line 225.

Point 4: Authors should provide the weight of PNF extract and D101 119 macroporous resin used for the enrichment.

Response 4: Thank you for your comments. The weight of PNF for enrichment is 9.85g and the weight of D101 macroporous resin is 197g have been added in 257 lines of the paper.

Point 5: Weight of total saponins from 75% aqueous ethanolic extract after 50% ethanolic elution should be mentioned.

Response 5: Thank you for your comments. The total saponins weight of the extract after 50% ethanol elution was added to Line 259 of the article, which was 2.73g, and the standard curve of the standard solution of ginsenoside Re in Figure 3 was added.

Point 6: There are still so many English typographical mistakes which needs to be taken care of.

Response 5: Thank you for your comments. The author has checked and corrected.